# Embryonic attenuated Wnt/β-catenin signaling defines niche location and long-term stem cell fate in hair follicle

**Zijian Xu[1,2], Wenjie Wang[2], Kaiju Jiang[2], Zhou Yu[2], Huanwei Huang[2], Fengchao Wang[2], Bin Zhou[3], Ting Chen[2]***

[1]College of Biological Sciences, China Agricultural University, Beijing, China; [2]National Institute of Biological Sciences, Beijing, China; [3]Key Laboratory of Nutrition and Metabolism, Institute for Nutritional Sciences, Shanghai Institutes for Biological Sciences, Graduate School of Chinese Academy of Sciences, Chinese Academy of Sciences, Shanghai, China

**Abstract** Long-term adult stem cells sustain tissue regeneration throughout the lifetime of an organism. They were hypothesized to originate from embryonic progenitor cells that acquire long-term self-renewal ability and multipotency at the end of organogenesis. The process through which this is achieved often remains unclear. Here, we discovered that long-term hair follicle stem cells arise from embryonic progenitor cells occupying a niche location that is defined by attenuated Wnt/β-catenin signaling. Hair follicle initiation is marked by placode formation, which depends on the activation of Wnt/β-catenin signaling. Soon afterwards, a region with attenuated Wnt/β-catenin signaling emerges in the upper follicle. Embryonic progenitor cells residing in this region gain expression of adult stem cell markers and become definitive long-term hair follicle stem cells at the end of organogenesis. Attenuation of Wnt/β-catenin signaling is a prerequisite for hair follicle stem cell specification because it suppresses Sox9, which is required for stem cell formation.

**\*For correspondence:** chenting@nibs.ac.cn

**Competing interests:** The authors declare that no competing interests exist.

## Introduction

Long-term adult stem cells (SCs) are defined by their ability to continuously generate all downstream differentiated cell lineages as well as regenerate themselves throughout the lifetime of an organism (*Fuchs and Chen, 2013*). Although rudimentary tissue progenitor cells in embryos give rise to all cells in adult tissue, these cells are not SCs, because they are destined to change identity as development proceeds. It has been postulated that definitive tissue SCs originate from tissue embryonic progenitor cells that acquire the capacity for long-term self-renewal and multipotency at the end of organogenesis (*Slack, 2008*). However, the underlying cellular and molecular mechanisms of these processes remain unknown. Deciphering the process leading to localized SC emergence during normal embryonic development will likely reveal principles that can be used to acquire and maintain the long-term self-renewal and regenerative potentials that are prerequisites for SC-based therapies.

Adult SCs in hair follicles (HFs) are located at the so-called 'outer bulge region'. The bulge region constitutes the bottom of the permanent portion of a HF. It contains two layers of epithelial cells: the CD34- inner layer niche cells and the CD34+ outer layer stem cells. Although HFs begin to develop in the embryo, the bulge structure is only formed when HFs enter the first postnatal telogen (resting phase); this delineates the end of organogenesis and the emergence of the adult SC niche (*Paus et al., 1999*; *Morris et al., 2004*; *Tumbar et al., 2004*; *Blanpain et al., 2004*; *Zhang et al., 2009*; *Hsu et al., 2011*; *Chen et al., 2012*).

**eLife digest** Many tissues and organs in an adult's body – including bone marrow, skin and intestines – contain a small number of cells called adult stem cells. These cells usually stay dormant within these tissues (at a site called a 'niche') until they are required to repair damaged or lost cells. At this point, adult stem cells can specialize, or 'differentiate', into the many different cell types that make up the tissue or organ where they reside.

The cells that produce hairs are an example of adult stem cells. In mammals, hairs grow from structures called hair follicles that are found in the skin, and over the life of an animal, old hairs are shed and replaced. Previous research had suggested that certain embryonic cells are set to become hair follicle stem cells before the hair follicles emerge in the adult tissue. However it remained unclear how this decision is made, and which genes and molecules are involved in this process.

Xu et al. have now found that, in mice, the fate of hair follicle stem cells is decided at an early stage in development, when the hair follicle is a simpler structure called a 'hair peg'. Cells near the upper part of the hair peg tend to become dormant and adopt an adult stem cell fate, while the ones in the lower part are more likely to differentiate straight away. This shows that the position, hence the niche environment, plays a key role in determining these different cells' fates.

Xu et al. went on to discover that the decision for a cell to become a hair follicle stem cell relies on reduced signaling through the so-called Wnt signal pathway. Understanding how adult stem cells become established during development may help future efforts to grow tissues and organs in the laboratory for research purposes or organ transplantation.

Initiation of HF organogenesis is marked by placode emergence. Activation of Wnt/β-catenin signaling in basal epithelial cells by locally expressed Wnt ligands is both necessary and sufficient to induce placode formation (*van Genderen et al., 1994*; *Gat et al., 1998*; *Huelsken et al., 2001*; *Andl et al., 2002*; *Jamora et al., 2003*; *Millar et al., 1999*; *Reddy et al., 2001*; *Fu and Hsu, 2013*). Concomitant with Wnt ligand production, placode epithelial cells also express the Wnt inhibitor Dkk (*Bazzi et al., 2007*). The combination of the short-range activator Wnt and the long-range inhibitor DKK function in a reaction-diffusion mechanism to suppress Wnt/β-catenin signaling in cells surrounding existing placodes, thereby regulating HF spacing (*Sick et al., 2006*). Contrary to the required activation of Wnt/β-catenin signaling, inhibition of BMP signaling through the dermal BMP inhibitor Noggin is necessary for placode morphogenesis (*Jamora et al., 2003*; *Botchkarev et al., 1999*; *Kobielak et al., 2003*).

After placode formation, signaling events downstream of Wnts/BMPs drive HF to further develop into the hair germ and then the hair peg. Shh is expressed by placode epithelial cells (*Oro et al., 1997*; *St-Jacques et al., 1998*; *Morgan et al., 1998*). In the absence of Shh, dermal condensate fails to aggregate and HF development arrests at the placode stage instead of developing into the hair germ. This implicates Shh in establishing proper epithelial-mesenchymal crosstalk (*St-Jacques et al., 1998*; *Oro and Higgins, 2003*; *Levy et al., 2007*). Guard hairs, as the first HFs to appear in the embryonic backskin, are uniquely dependent on Eda/EdaR signaling (*Laurikkala et al., 2002*; *Mustonen et al., 2004*). Eda, which is itself a Wnt signaling target (*Laurikkala et al., 2001*), can induce the expression of BMP inhibitors and *Shh* in guard hair (*Pummila et al., 2007*).

Placode progenitor cells generate all cells in adult HFs (*Levy et al., 2007*). Some of their progeny cease further development at a particular point and become definitive adult HFSCs. Previous studies using label-retaining methods demonstrated that putative HFSCs are present in postnatal developing HFs as slow cycling cells, and it was shown that their specification requires the transcription factor Sox9 (*Nowak et al., 2008*). Intriguingly, placode cells already express adult HFSC markers such as Lhx2 and Sox9, although in a largely non-overlapping pattern. Another HFSC marker, Nfatc1, appears in the subsequent hair peg (*Rhee et al., 2006*; *Horsley et al., 2008*; *Vidal et al., 2005*). These dynamic expression patterns suggest that cells in the placode and hair peg are heterogeneous. However, whether or not HFSC fate is already pre-determined at these early developmental stages is not clear. Other critical unanswered questions include the following: Are adult HFSCs remnant of embryonic progenitor cells that maintain their embryonic developmental potential, or do

they, alternatively, come from progenitor cells that gain long-term potential? What are the underlying mechanisms? What determines the niche location and where do HFSCs become localized? The current study addresses these key questions.

## Results

### The embryonic cellular origin of adult hair follicle stem cells

To uncover the cellular origin of HFSCs and to identify the time point of their specification, it will first be necessary to perform lineage-tracing experiments. These can be done by labelling distinct cell populations at the rudimentary stages and later determining whether SCs come from these initially labelled progenitor cells (*Figure 1A*). We chose tail skin HFs for this study. Unlike un-patterned back skin HFs (*Figure 1—figure supplement 1A*), tail skin HFs are arranged in triplets, and the growth of two secondary outer follicles is typically initiated next to a primary central follicle after it has already developed (*Figure 1—figure supplement 1B*). By inducing Cre activation at specific time points and focusing on HFs in a chosen area, we can label progenitor cells in defined developmental stages and continue to follow their fates in individual HFs until the end of organogenesis, when the bulge forms (*Figure 1B,C*; *Figure 1—figure supplement 2A-C*).

To test whether distinct HFSC precursors are already present at the initial developmental stages, we used several CreER knock-in lines that can label different cell populations of the placode, hair germ, and hair peg. *Shh* starts to be expressed in the placode (*St-Jacques et al., 1998*). When we used inducible *Shh-CreER::Rosa-stop-mTmG* (*Harfe et al., 2004*) mice to label *Shh*-expressing cells in either the placode or the hair germ, we found that in both cases *Shh+* progenitors cells can become HFSCs (*Figure 1D*). Unexpectedly, *Shh*-expressing hair peg cells lost the ability to form HFSCs and mostly contributed to differentiated lineages (*Figure 1D,E*). Since *Shh*-expressing placode cells generate the majority of cells in hair peg, including those still expressing *Shh*, this intriguing change in cell fate indicates that progenies of *Shh*-expressing placode cells separate into two different populations: the population still expressing *Shh* loose the potential to form HFSCs, while the cells with down regulated *Shh* expression in hair peg can presumably still become HFSCs.

Consistent with this hypothesis, two adult HFSC markers, *Lgr5* and *Nfatc1,* first begin to be expressed in hair peg, with complementary patterns to that of *Shh* (*Figure 1D*). Using *Nfatc1-CreER::Rosa-stop-mTmG* mice (*Tian et al., 2014*), we found that *Nfatc1*-expressing cells first appear at the upper hair peg and remain relatively dormant there, without actively contributing to HF down growth (*Figure 1F*). Then, at the end of organogenesis, *Nfatc1*-expressing hair peg cells become the sebaceous gland above bulge and most of the HFSCs (*Figure 1D*). A second HFSC marker, Lgr5, follows a similar pattern, albeit with different cell fate preferences. Using *Lgr5-GFP-CreER:: Rosa-stop-tdTomato* mice (*Jaks et al., 2008*), we found that *Lgr5*-expressing cells appear first in the middle portion of the hair peg. They eventually become HFSCs in the lower bulge and secondary hair germ cells below the bulge (*Figure 1D*). The labeling efficiencies of the CreER lines used here were very high; almost all the HFs in the chosen region targeted for quantifications were labeled, and we did not detect any leaky induction in the absence of Tamoxifen injection (*Figure 1—figure supplement 2D-G*).

Collectively, these results suggest that HFSCs originate from progenitor cells in hair peg that lose *Shh* expression but gain the expression of adult stem cell markers (*Figure 1G*). The hair peg SC precursors are distinguished from the broader placode and hair germ cells in that they stop further development once they are specified. During subsequent HF down growth, these cells remain dormant in their original position until the end of organogenesis. At that point, they incorporate into the outer layer of the bulge and become HFSCs. This leads to an interesting question: is the reason these cells gain expression of stem cell markers and become dormant due to signals that they receive at the specific upper follicle position?

### Embryonic niche induces hair follicle stem cell fate

We used a two-photon laser to precisely ablate SC precursors in hair peg labeled by *Nfatc1-CreER:: Rosa-stop-tdTomato::K14H2BGFP* (*Figure 2A,B*). Successful and precise cell ablation was achieved without affecting normal development of neighboring HFs (*Figure 2—figure supplement 1A*) (*Rompolas et al., 2012*). Following complete ablation of *Nfatc1-CreER*-labeled cells, we observed

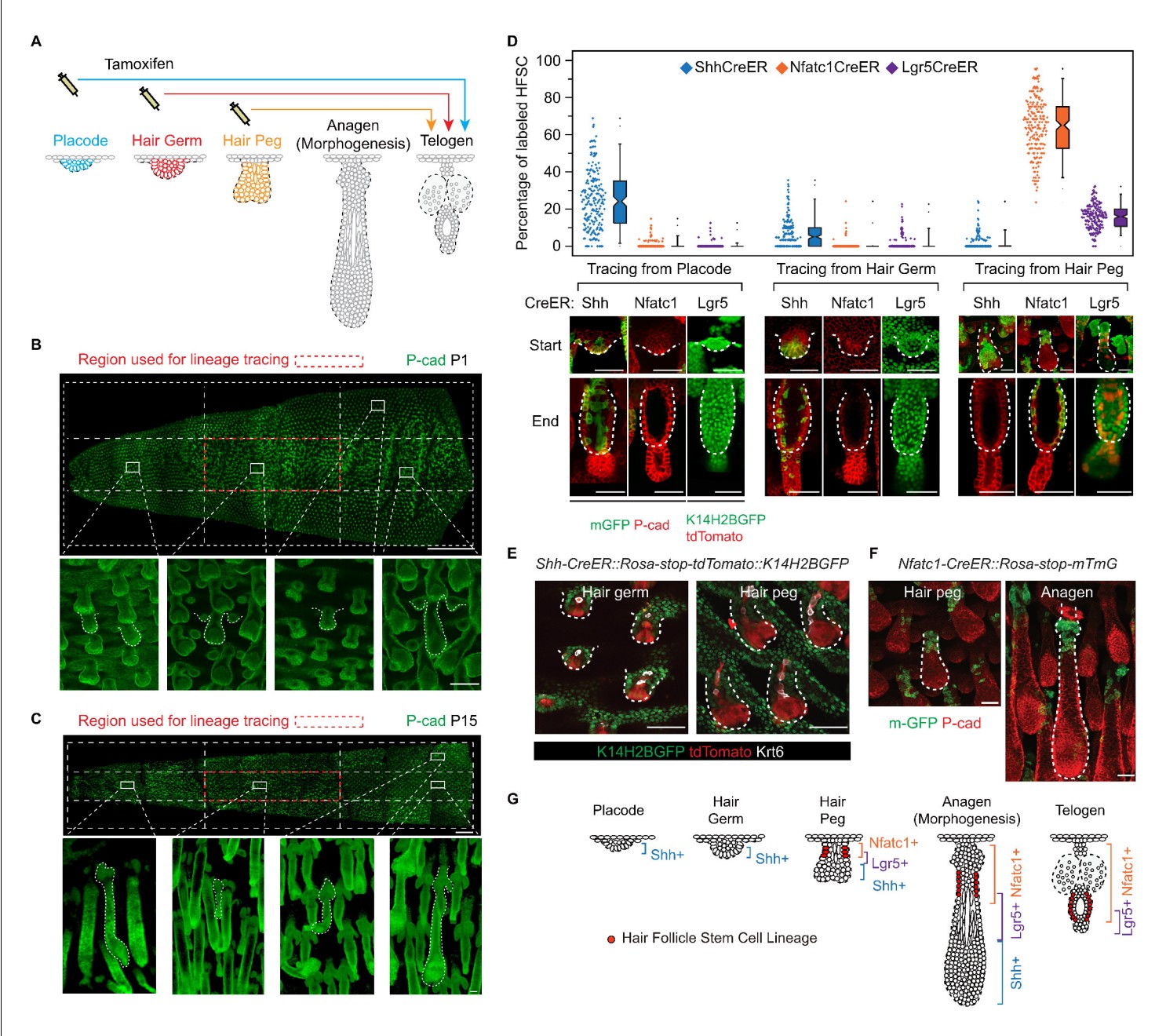

**Figure 1.** Embryonic cellular origin of adult hair follicle stem cells. (**A**) Diagram of hair follicle morphogenesis and the lineage-tracing experiment. All lineage-tracing experiments ended at the first telogen, but started at different stages including the placode, hair germ, and hair peg stages. (**B,C**) Representative images of tail skin hair follicle organogenesis. Red boxes indicate the regions used for quantification in the lineage-tracing experiments. The hair cycle in tail skin progresses along the anterior to posterior and in the dorsal to ventral directions. At postnatal day 1 (P1), in the chosen region, the primary central hair follicles are in the hair peg stage while the secondary outer follicles are in the placode stage. At P15, in the chosen region, primary central hair follicles are in the telogen phase. Scale bar: 1500 μm for the whole mount image; 100 μm for the enlarged images. (**D**) Summary of the lineage-tracing experiments with: *Shh-CreER::Rosa-stop-mTmG, Nfatc1-CreER::Rosa-stop-mTmG*, and *Lgr5-GFP-CreER::Rosa-stop-tdTomato:: K14H2BGFP* mice. Percentages of labelled HFSCs at the telogen phase were quantified using whole mount sample images. Representative images are single confocal Z slices from the data used for quantification. N=5 mice, >200 HFs. Raw data are plotted to the left of each box-and-whisker plot: the median and the 25th and the 75th percentiles are denoted by notches and the bottom and top boxes, respectively; the 5th and 95th percentiles are denoted as whiskers. Scale bar: 50 μm. (**E**) In both the hair germ and the hair peg, the progenies of Shh+ cells in the center of HFs express the differentiation marker Keratine 6 (Krt6). Scale bar: 50 μm. (**F**) Lineage-tracing experiment starting at the hair peg stage and ending at morphogenesis

*Figure 1 continued on next page*

Figure 1 continued

anagen (growth phase) using *Nfatc1-CreER::Rosa-stop-mTmG* mice. Note that the labeled cells stay dormant at their original position without contributing to HF down growth. Scale bar: 50 μm. (G) Model depicting the spatial and temporal pattern of hair follicle stem cell emergence.

The following figure supplements are available for figure 1:

**Figure supplement 1.** Difference between back and tail skin hair follicle organogenesis.

**Figure supplement 2.** Diagram, quantification, and specificity of CreER lines for the lineage-tracing experiments.

de novo bulge formation at the end of organogenesis (*Figure 2C*). The newly formed bulge even had the same number of SCs as the control HFs (*Figure 2D*). To achieve maximum labeling and ablation efficiency of *Nfatc1+* cells, we conducted the ablation experiment only in HFs along the anterior-posterior midline of the dorsal tail skin, as this region has the highest Tamoxifen induction efficiency (*Figure 2—figure supplement 1B*). Under this condition, almost all SCs in neighboring control HFs were tdTomato positive, while the de novo formed SCs in the ablated HFs were all tdTomato negative (*Figure 2E*). Importantly, the de novo formed HFSCs were functional as indicated by their ability to support HF down growth (*Figure 2F-I* and *Figure 2—figure supplement 1C*). These results indicate that niche location in hair peg determines HFSC fate.

## Attenuated Wnt/β-catenin signaling is uniquely associated with the embryonic niche

To understand the underlying mechanism leading to niche-induced HFSC specification, we profiled the expression of mRNA isolated from purified hair peg cells expressing *Shh* or *Nfatc1* (*Figure 3A-C*). For the purpose of identifying niche-defining factors, we focused on genes associated with extracellular signal paths. Rather than identifying signals that were uniquely present in hair peg niche occupying cells (*Nfatc1+* cells), we found signals that were uniquely absent from these cells. Based on DAVID functional gene annotation analysis (*Dennis et al., 2003*), among genes that were expressed ≥2-fold more in *Shh+* cells than in *Nfatc1+* cells, we found that genes of the Wnt signaling pathway were enriched prominently (*Figure 3D*). Several canonical Wnts and some well-known Wnt/β-catenin target genes, including *Axin2*, were highly expressed in *Shh+* cells (*Figure 3—figure supplement 1*, *Figure 3—source data 1*). Using either real-time PCR or in situ hybridization, we confirmed that *Axin2, Wnt10b, Wnt3,* and *Lef1* were all expressed at higher levels in *Shh+* cells than in *Nfatc1+* hair peg cells (*Figure 3E-G*). We then used *TOPGAL* Wnt/β-catenin signal reporter mice to validate the patterns we had observed. Starting in the placode stage and persisting through to subsequent development stages, *Shh+* cells showed consistent co-localization with β-gal+ cells (*Figure 3H*). These results suggest that the hair peg niche position is associated with an absence of Wnt/β-catenin signaling.

To directly test whether or not attenuated Wnt/β-catenin signaling in epithelial cells was correlated with HFSC specification at the hair peg stage, we conducted experiments with *Axin2-CreER::Rosa-stop-mTmG* mice (*Lim et al., 2013*). As expected, Wnt/β-catenin signal responsive cells in placode give rise to most of the HFSCs (*Figure 4A,B*) (*Huelsken et al., 2001*). This clear correlation at the placode stage was dramatically decreased at the hair germ stage, and even further decreased at the hair peg stage. Only a very small proportion of HFSCs came from *Axin2+* hair peg cells; the majority of HFSCs came from *Axin2-* hair peg cells (*Figure 4A,B*). This suggests that from the placode stage to the hair peg stage, cells that maintain active Wnt/β-catenin signaling have dramatically decreased potential to form HFSC.

During HF down growth, several signaling pathways are known to be involved in promoting the differentiation or changes in cell fates (*Millar, 2002*). Although the Wnt/β-catenin signaling pathway is the only such pathway identified from our RNA-seq results, we checked to see if similar associations existed for other pathways. We used *Gli1-CreER::Rosa-stop-mTmG* mice to detect descendents of Shh signal-responsive cells and *Id2-CreER::Rosa-stop-mTmG* mice to detect descendents of BMP signal-responsive cells (*Ahn and Joyner, 2004*; *Rawlins et al., 2009*). Almost all the HF cells were *Gli1+* in all the early developmental stages examined (*Figure 4C*). *Id2* only became active in terminally differentiated hair bulb cells after future HFSC fate had been specified (*Figure 4D*). The

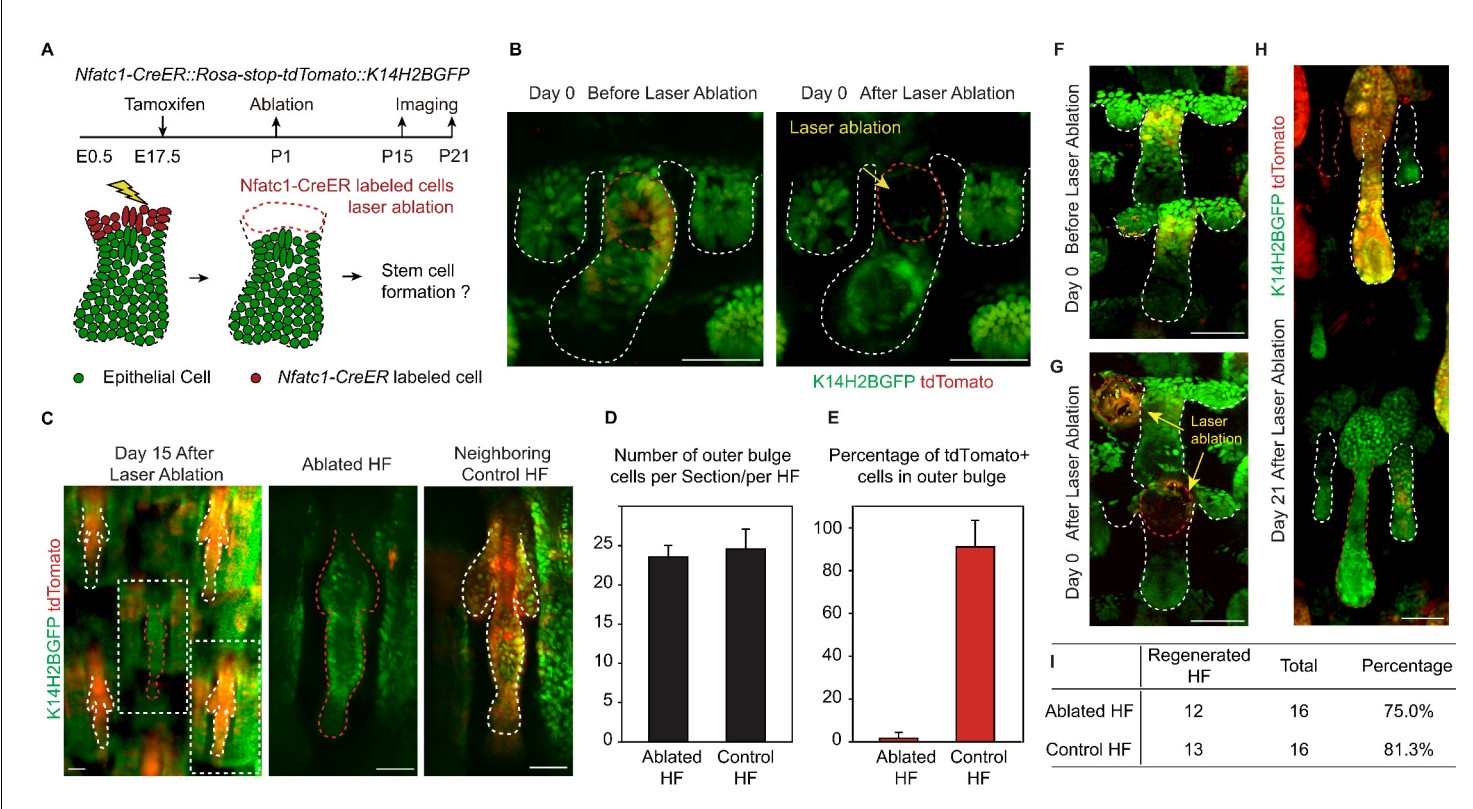

**Figure 2.** Niche position in the hair peg determines hair follicle stem cell fate. (**A**) Diagram of two-photon-mediated cell ablation experiment using *Nfatc1-CreER::Rosa-stop-tdTomator::K14H2BGFP* mice. (**B**) Representative images of hair follicles before and after cell ablation in a live animal. (**C**) Representative whole-mount images of hair follicles 15 days after cell ablation. Notice that the control hair follicle has tdTomato+ cells in the bulge while the ablated hair follicle has a normal bulge composed of tdTomato- cells. (**D**) Quantification of hair follicle stem cell number at telogen using whole mount samples. N=6 mice, >12 HFs. (**E**) Quantification of percentage of tdTomato+ outer bulge cells at telogen using whole mount samples. N=6 mice, >12 HFs. (**F–H**) Representative images of hair follicles before (**F**), immediately after (**G**), and 21 days after (**H**) cell ablation from the same mouse. Notice that both the control hair follicle and ablated hair follicle enter anagen. (**I**) Quantification of hair follicles that have started regeneration 21 days post cell ablation. N=5. Scale bars: 50 μm

The following figure supplement is available for figure 2:

**Figure supplement 1.** Ablation specificity, Tamoxifen induction efficiency, and whole-mount views of the two-photon-mediated cell ablation experiments.

results from these experiments indicate that attenuated Wnt/β-catenin signaling is uniquely associated with the HFSC-inducing niche location in hair peg (*Figure 4E*).

## Elevated Wnt/β-catenin signaling abolishes HFSC specification and suppresses *Sox9* expression

To investigate whether the observed correlation of attenuated Wnt/β-catenin signaling with HFSC specification is a functional requirement, we increased Wnt/β-catenin signaling in cells occupying hair peg niche position to see if that would affect HFSC specification. This was achieved by using *Nfatc1-CreER::Exon3-Ctnnb1^{fl/wt}* mice (*Harada et al., 1999*). Activation of canonical Wnt/β-catenin signaling in *Nfatc1+* cells was confirmed by nuclear β-catenin staining (*Figure 5A,B*). Initial HF development was normal in *exon3-Ctnnb1* heterozygous (Het) mice. However, after morphogenesis, rather than entering a new hair cycle like the WT, the hair shafts of Het mice were shed (*Figure 5— figure supplement 1A,B*). Section staining revealed that bulge formation was completely abolished in *exon3-Ctnnb1* Het mice. There were disorganized Keratin6+ cells where the bulge should have been, but there were no cells expressing the HFSC markers CD34 or Sox9 (*Figure 5C-E*). Thus, when

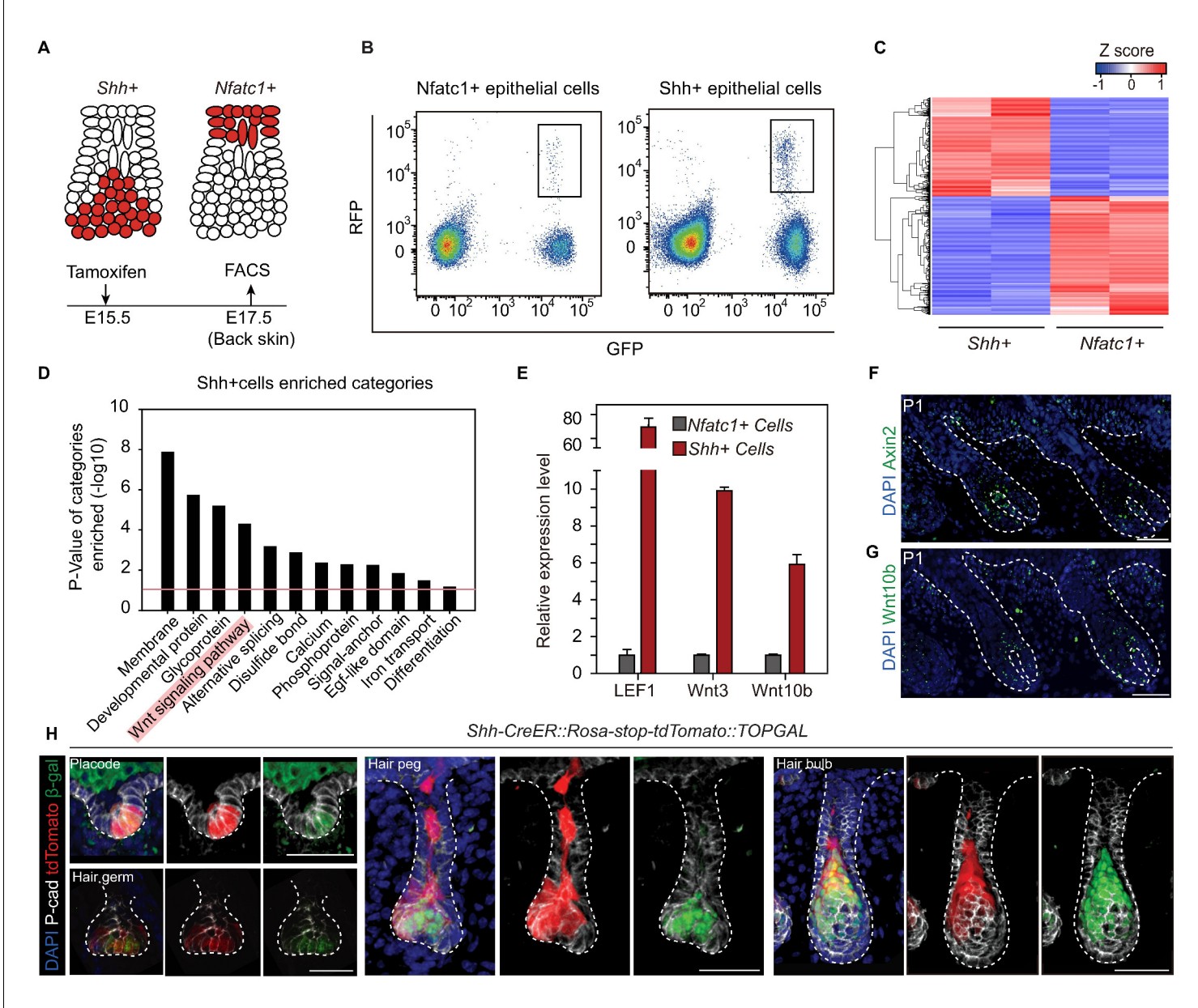

**Figure 3.** Unbiased RNA-seq analysis reveals factors that define the hair peg niche. (**A**) Diagram of the FACS experiments using *Nfatc1- and Shh-CreER::Rosa-stop tdTomato::K14H2BGFP mice*. (**B**) FACS isolation of distinct GFP+RFP+ populations to obtain *Shh+* and *Nfatc1+* epithelial cells. (**C**) Unsupervised hierarchical clustering and heat map display of genes that were differentially expressed between *Shh+* cells and *Nfatc1+* cells. N=2 (**D**) Gene Ontology analysis of ≥2-fold up-regulated genes in *Shh+* cells compared to *Nfatc1+* cells. The Wnt signaling pathway is highlighted. (**E**) Validation of differentially expressed genes using qPCR. N=3. (**F–G**) In situ staining of *Axin2* (**F**) and *Wnt10b* (**G**) in developing hair follicles. (**H**) *Shh+* cells are Wnt/β-catenin signal responsive cells. *Shh* expression was represented by tdTomato in *Shh-CreER::Rosa-stop-tdTomato* mice. Wnt/β-catenin-responsive cells were detected by β-gal staining in *TOPGAL* mice. Scale bars: 50 μm.

The following source data and figure supplement are available for figure 3:

**Source data 1.** RNA-seq results of differentially expressed genes between Nfatc1+ and Shh+ cells.

**Figure supplement 1.** RNA seq results of representative genes from different populations.

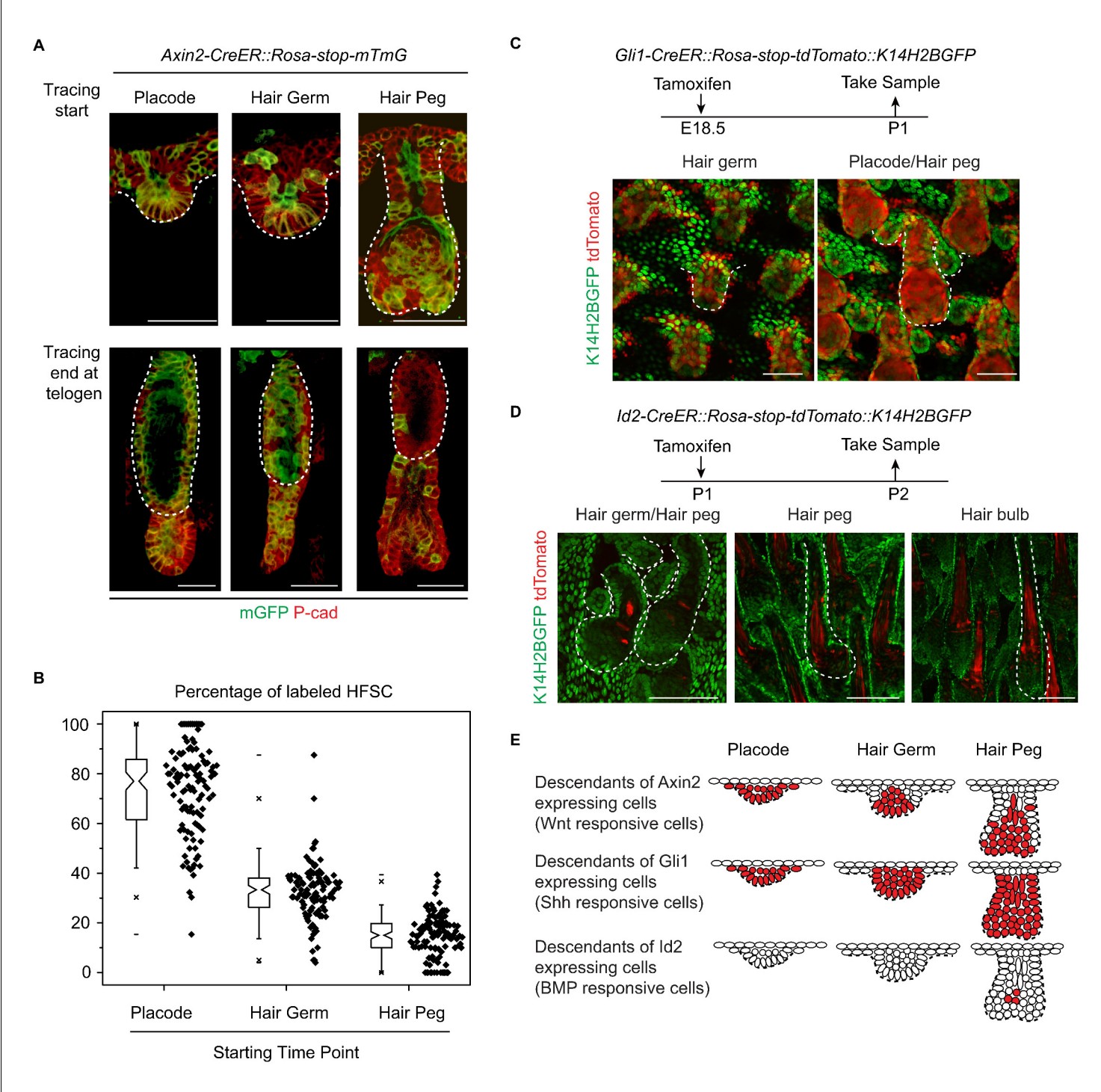

**Figure 4.** Attenuated Wnt/β-catenin signaling is uniquely associated with hair follicle stem cell specification. (A–B) Lineage tracing with *Axin2CreER:: Rosa-stop-mTmG* mice. N=5 mice, >120 HFs. (C) Descendants of Gli1+ cells were represented by the expression of tdTomato in *Gli1-CreER::Rosa-stop-tdTomato::K14H2BGFP* mice. Note that Gli1+ descendants are located in almost all the hair follicle cells at the placode, hair germ, and hair peg stages. (D) Descendants of Id2+ cells were represented by the expression of tdTomato in *Id2-CreER::Rosa-stop-tdTomato::K14H2BGFP* mice. Note that Id2+ descendants were only located at the hair bulb stage matrix and pre-cortex area. (E) Diagram summarizing the descendants of the Wnt-, Shh-, and BMP-signal responsive cells at different early developmental stages. Scale bars: 50 μm.

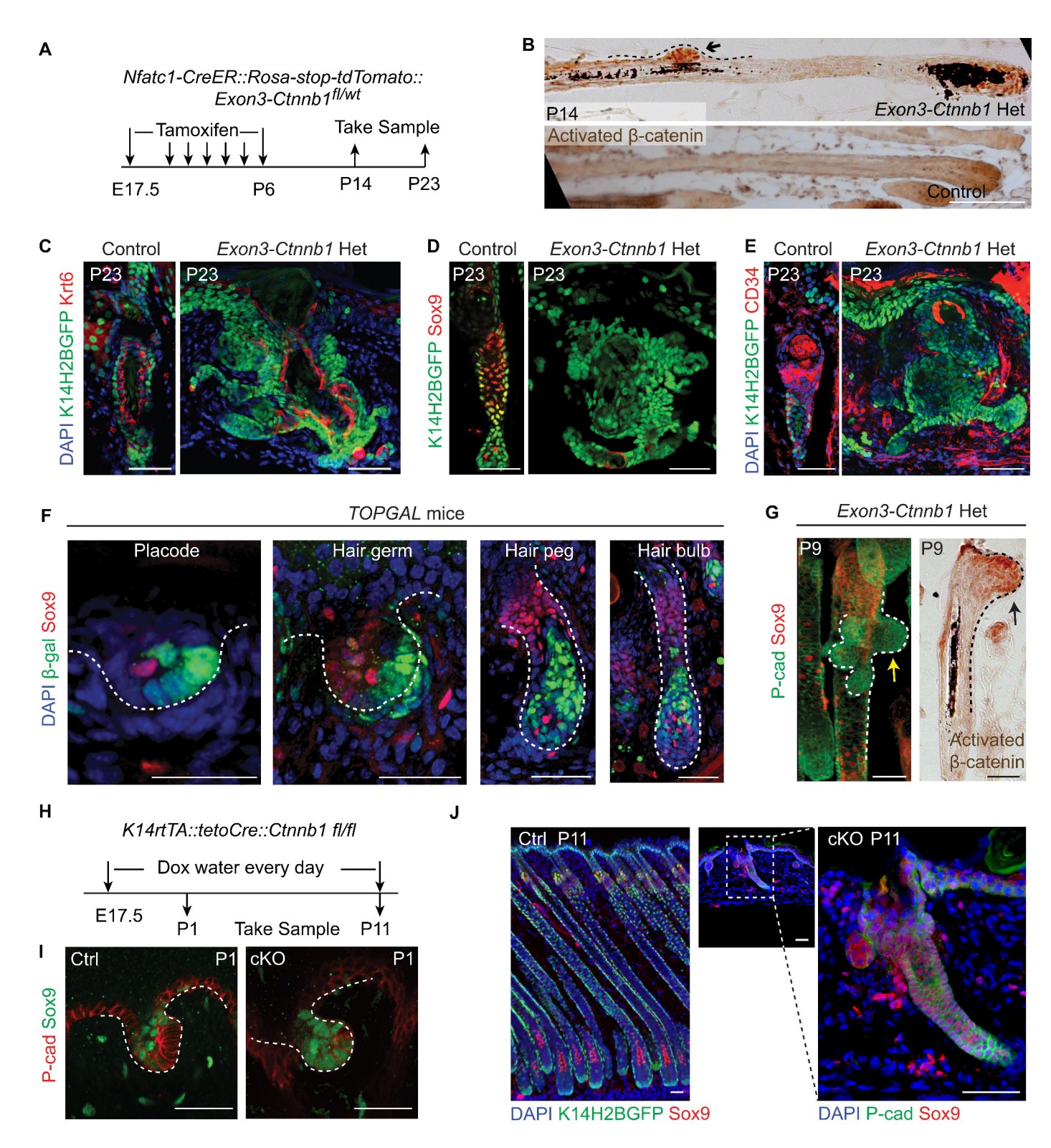

**Figure 5.** Elevated Wnt/β-catenin signaling abolishes hair follicle stem cell specification and suppresses Sox9 expression in hair follicles. (**A**) Diagram of the experiments using *Nfatc1-CreER::Rosa-stop-tdTomato::Exon3-Ctnnb1*^fl/wt^ mice. (**B**) Nuclear β-catenin staining indicates successful activation of Wnt/β-catenin signaling in upper hair follicle. (**C–E**) Abolished bulge niche formation and hair follicle stem cell specification in *exon3-Ctnnb1* Het HFs

*Figure 5 continued on next page*

*Figure 5 continued*

compared to WT HFs. Krt6 (C) is a marker for inner layer bulge cells serving as a niche that can maintain quiescence for outer layer HFSCs. Sox9 (D) and CD34 (E) are adult hair follicle stem cell markers. (F) Wnt/β-catenin signal responsive cells, represented by β-gal positive cells in *TOPGAL* mice, do not express *Sox9* during normal development. (G) Activation of Wnt/β-catenin signaling suppresses *Sox9* expression in vivo. The arrows point to protrusions resulting from elevated Wnt/β-catenin signaling in *exon3-Ctnnb1* Het HFs. (H) Diagram of the experiments using *K14-rtTA::teto-Cre::Ctnnb1$^{fl/fl}$* mice. β-catenin is conditionally deleted in epithelial cells by feeding mice with Doxycyclin from E17.5 to P11. Samples were taken at P1 and P11. (I–J) Loss of Wnt/β-catenin signaling leads to expanded *Sox9* expression in HFs at both the hair germ stage (I) and in postnatal skin (J). Scale bars: 50 μm.

The following figure supplement is available for figure 5:

**Figure supplement 1.** Effects of both gain and loss of function studies targeting Wnt/*β*-catenin signaling in skin.

Wnt/β-catenin signaling was increased in hair peg niche-occupying cells, neither HFSCs nor the bulge niche formed at the end of organogenesis.

Interestingly, with increased time, the *exon3-Ctnnb1* Het mice started to generate new hair coats with normal bulges and HFSCs that expressed CD34 and Sox9 (*Figure 5—figure supplement 1A-C*). Cells expressing *exon3-Ctnnb1* have spontaneously disappeared from HFs, and the remaining mosaic WT cells organized into a de novo niche after entering the normal hair cycle process (*Figure 5—figure supplement 1D*). These observation further supports our conclusion that attenuated Wnt/β-catenin signaling is required for HFSC fate specification.

It is intriguing that Sox9 expression is not present in cells with elevated Wnt/β-catenin signaling. Sox9 is known to be an essential intrinsic factor that is required for HFSC formation. Ablation of epithelial *Sox9* completely blocks HFSC specification (*Nowak et al., 2008*). To determine if the canonical Wnt signal suppresses Sox9 expression in HFs, first we observed that in normal developing HFs, Sox9 expressing cells and Wnt/β-catenin signal responsive cells are mutually exclusive (*Figure 5F*). Second, soon after activation of Wnt/β-catenin signaling in *Nfatc1+* cells, we observed small protrusions from HFs that were positive for nuclear β-catenin. Sox9 expression in these small protrusions was completely absent, while cells immediately next to these protrusions express Sox9 (*Figure 5G*). Lastly, in the Wnt/β-catenin loss of function mutant (*K14-rtTA::teto-Cre::Ctnnb1$^{fl/fl}$*) (*Nguyen et al., 2006*; *Perl et al., 2002*), HF development is defective but Sox9 expression is expanded to the extent that all remaining cells in HFs express Sox9 (*Figure 5H-J*). These results suggest that the Wnt/β-catenin signaling suppresses Sox9 expression in HFs. Given that the loss of Sox9 expression completely blocks HFSC specification, the requirement for attenuation of Wnt/β-catenin signaling as a prerequisite for HFSC specification is related to enabling Sox9 expression. It is noteworthy that hair shaft formation is aborted in *K14-rtTA::teto-Cre::Ctnnb1$^{fl/fl}$* mice (*Figure 5—figure supplement 1E,F*). This indicates that epithelial Wnt/β-catenin signaling is required for progenitor cell differentiation, and may further explain why activation of Wnt/β-catenin signaling prevents HFSC formation.

## Embryonic Wnt/β-catenin signaling decreases long-term potential of hair follicle stem cells

Given that attenuated Wnt/β-catenin signaling is required for HFSC specification, it was puzzling to observe that a small portion of HFSCs originated from *Axin2+* hair peg cells (*Figure 4A,B*). This suggests that there are two origins of HFSCs: a small portion of HFSCs come from hair peg cells maintaining active Wnt/β-catenin signaling, while the majority HFSCs come from progenitor cells with attenuated Wnt/β-catenin signaling in hair peg. To investigate the long-term consequences of this heterogeneity, we followed the fates of HFSCs from the two different origins through multiple hair cycles; the Wnt/β-catenin signal positive origin was represented by the *Axin+* lineage while the Wnt/β-catenin signal negative origin was represented by the *Sox9+* lineage (*Figure 6A*).

Consistent with results presented in *Figure 4A*, ~15% of HFSCs in the first telogen originated from *Axin2+* hair peg cells. In the first growth phase (anagen) following the first telogen, these HFSCs contributed to both outer root sheath (ORS) cells and to matrix differentiating cells (*Figure 6B*). Of note, after two full hair cycles, in the third telogen, *Axin2+* hair peg cell formed HFSCs have been mostly exhausted (*Figure 6C*). On the contrary, *Sox9+* hair peg precursor cells were the origin of most of the HFSCs at the first telogen. During the first anagen, these HFSCs contributed preferentially to ORS rather than to the differentiated lineages in the matrix. Instead of

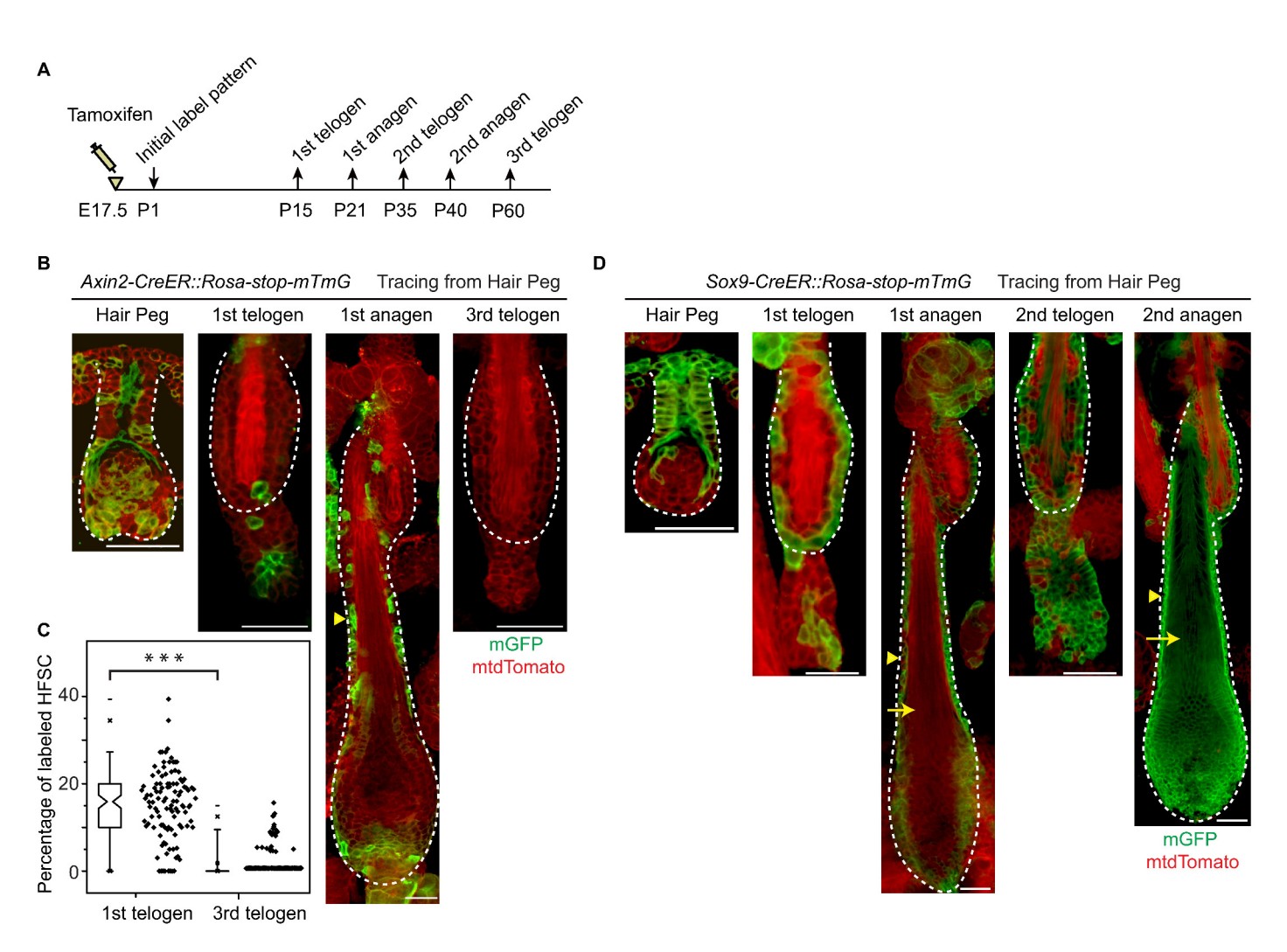

**Figure 6.** Embryonic Wnt/β-catenin signaling diminishes the long-term self-renewal ability of hair follicle stem cells in adult. (**A**) Diagram depicting the time points of the lineage tracing experiment. (**B–C**) Long-term lineage tracing experiments using *Axin2-CreER::Rosa-stop-mTmG* mice. Note the disappearance of labelled HFSCs in the third telogen. Representative images shown are single confocal Z slice from the data used for quantification. N=3 mice, >110 HFs. ***p<0.0001. (**D**) Long-term lineage tracing experiments with *Sox9-CreER::Rosa-stop-mTmG* mice. Arrowheads indicate ORS and arrows indicate terminally differentiated layers. Note the change in downstream progeny fates of labeled cells at different hair cycles. Scale bars: 50 μm.

The following figure supplement is available for figure 6:

**Figure supplement 1.** Long term cell fate of Lgr5-CreER labelled hair peg cells.

decreasing, HFSCs originated from *Sox9+* hair peg cells persisted and expanded after one round of hair cycle to become essentially all of the HFSCs in the bulge and most of the secondary hair germ cells below the bulge. For the second anagen, the labeled HFSCs generate all of the downstream lineages including, ORS, matrix, and hair shaft cells (*Figure 6D*).

These results suggest that embryonic Wnt/β-catenin signaling diminishes the long-term potential of HFSCs. We have shown that Wnt/β-catenin signaling is required for terminal differentiation of the hair shaft. Also it suppresses Sox9 expression, which is important for HFSC maintenance through inhibition of differentiation (*Kadaja et al., 2014*). So it is possible that embryonic exposure to Wnt/β-catenin signaling primes the embryonic cells for differentiation via both positive induction and lack of Sox9 expression. This ultimately results in diminished long-term potential of the HFSCs.

## Discussion

### Hierarchical events leading to long-term hair follicle stem cell emergence

Our study elucidates the cascade of events that lead to long-term SC emergence in HFs during organogenesis (*Figure 7*). Wnt/β-catenin signaling is necessary for HF initiation (*Andl et al., 2002*; *Ito et al., 2007*). Early placode cells are Wnt/β-catenin signal responsive cells, and they generate all of the cells in adult HFs. At the hair peg stage, a localized Wnt/β-catenin signaling free zone emerges in upper HF. Progenitor cells residing in this embryonic niche position stop further development and become HFSCs at the end of organogenesis. These established HFSC precursor cells are not just remnants of embryonic progenitor cells, because they gain the expression of adult HFSC markers including *Nfatc1*. Embryonic cells residing in this Wnt/β-catenin signaling free zone, together with a few other cells located below them that have active Wnt/β-catenin signaling, give rise to the adult HFSC pool. However, only the HFSCs derived from Wnt/β-catenin signaling negative precursor cells have long-term self-renewal ability and can continuously support regeneration. The small portion of HFSCs that are derived from embryonic cells with active Wnt/β-catenin signaling are highly prone to differentiation as the hair cycle progresses and are soon exhausted after regeneration. So, in embryos, the attenuation of Wnt/β-catenin signaling not only defines the adult niche position and HFSC fate, but also has a long lasting effect on the subsequent fate choice and long-term potential of cells in adults.

It's noteworthy that we observed largely none overlapping labeling patterns of Lgr5+ and Axin2+ cells in hair peg (*Figure 6B*, *Figure 6—figure supplement 1B*). Also, Lgr5+ hair peg cells contributed HFSCs persist after multiple hair cycles instead of completely diminishing like Axin2+ cells contributed HFSCs (*Figure 6—figure supplement 1*). These results indicate it is possible that in hair follicle Lgr5 expression either reflects very low level of Wnt signaling, or it is induced by a different set of Wnt ligands than those induce Axin2.

### Restricted induction and maintenance of SC gene expression

Although we were able to pinpoint the main mechanism through which Wnt/β-catenin signaling blocks long-term HFSC formation, i.e. the suppression of *Sox9* expression, we suspect that this might not be the only mechanism involved in this process. It has been reported previously that *Sox9* expression is induced by Shh signaling (*Vidal et al., 2005*). Given that we observed extended activity of Shh signaling throughout follicle epithelial cells and observed only localized activity of Wnt/β-catenin signaling in lower follicle cells, we can explain the mutually exclusive pattern of *Sox9+* and Wnt/β-catenin signaling responsive cells. As development proceeds, the Sox9 expression pattern

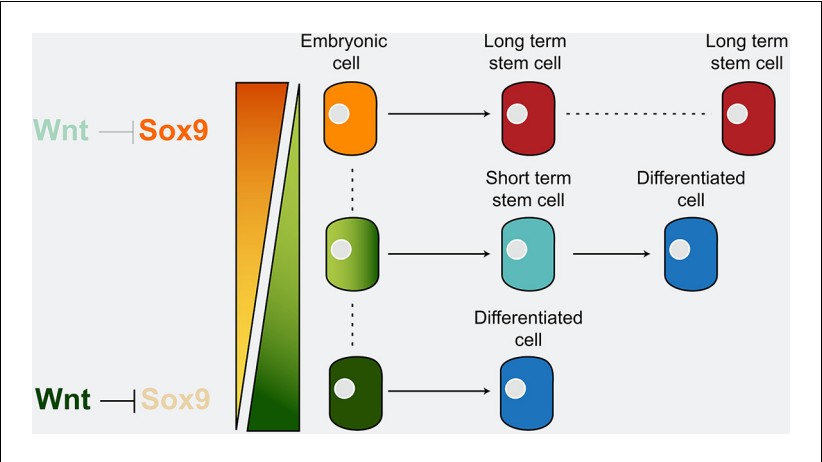

**Figure 7.** Model illustrating that long-term hair follicle stem cell emergence results from progenitors occupying an embryonic niche location, which is defined by the absence of Wnt/*β*-catenin signaling that would otherwise block the expression of a key factor required for stem cell specification.

continues to expand and become wider than the actual future bulge region; for example, Sox9 is expressed in most of the ORS cells. So, following the emergence of the Wnt/β-catenin signaling free zone in the hair peg, there are probably other mechanisms that help maintain the restricted niche position. One observation further supporting this hypothesis is that, once initiated in hair peg, the expression of *Nfatc1* is restricted to the upper HF where the future sebaceous gland and bulge will be; *Nfatc1* expression does not expand to the broader ORS like *Sox9* expression does. Nfatc1 was reported to be a target of BMP signaling in adult HFs (*Horsley et al., 2008*). However, we did not observe active BMP signaling at the hair peg niche location using a BMP reporter, a result similar to a published staining pattern of pSMAD1/5 in developing HFs (*Kandyba et al., 2014*). So, it remains to be determined which, if any, other mechanisms lead to the maintenance of a more restricted niche region that is reflected by the localized expression of stem cell markers.

## Mechanism leading to the emergence of a localized Wnt/β-catenin signaling free zone in the hair peg

Since attenuated Wnt/β-catenin signaling defines the niche location and long-term HFSC fate, the process of exactly how the Wnt/β-catenin signaling free zone emerges is of great interest. We observed increased expression of Wnt inhibitors such as *SFRP*s in *Nfatc1+* hair peg cells as compared to *Shh+* hair peg cells. However, given that HFSC precursor cells can be replaced by embryonic cells occupying the same hair peg niche position, it is unlikely that Wnt inhibitors expressed by the precursor cells themselves are the main causes of attenuated Wnt/β-catenin signaling in that position. Our RNA seq analysis revealed enrichment of *Wnt* and *Dkk* expression in Wnt/β-catenin signal positive *Shh+* hair peg cells as compared to *Nfatc1+* hair peg cells. We hypothesize that the combination of the lack of Wnt ligands, which are short-range signal molecules, and the presence of Wnt inhibitors such as DKK4, which are long-range signal molecules, at the embryonic niche location is the main reason contributes to the emergence of a Wnt/β-catenin signaling free zone in upper hair peg. A similar model has been used to explain HF density determination and epidermal SC regulation (*Sick et al., 2006*; *Lim et al., 2013*). To directly test the suppositions of this hypothesis, the protein localization patterns of Wnt and DKK will need to be dissected.

## Materials and methods

### Mice

*Nfatc1-CreER* mice were generated and provided by Dr. Bin Zhou (*Tian et al., 2014*). *K14-H2BGFP* mice were kindly provided by Dr. Elaine Fuchs. The *Shh-CreER* (*Harfe et al., 2004*), *Lgr5-GFP-CreER* (*Barker et al., 2007*), *Axin2-CreER* (*Lim et al., 2013*), *Id2-CreER* (*Rawlins et al., 2009*), *Gli1-CreER* (*Ahn and Joyner, 2004*), *Rosa-stop-mTmG* (*Muzumdar et al., 2007*), *Rosa-stop-tdTomato* (*Madisen et al., 2010*), *TOPGAL* (*DasGupta and Fuchs, 1999*), *Ctnnb1*<sup>fl/fl</sup> (*Huelsken et al., 2001*), *Exon3-Ctnnb1*<sup>fl/fl</sup> (*Harada et al., 1999*), *K14-rtTA* (*Nguyen et al., 2006*), and *teto-Cre* (*Perl et al., 2002*) mice have all been described previously. We generated the *Sox9-CreER* mice by integrating IRES-CreERT2-SV40pA cassettes into the 3′ UTR of the endogenous mouse *Sox9* gene, before the <u>aacatggaggacgattggagaatc</u> sequence via Cas9/RNA mediated gene targeting in zygotes. To perform the lineage tracing experiments, female *Rosa-reporter* mice were mated with male mice of the following genotypes: *Shh-, Lgr5-GFP-, Nfatc1-, Axin2-, Gli1-, Sox9-,* and *Id2-CreER*.

For the calculation of embryonic time points in the timed pregnancy experiments, the morning of the vaginal plug date was designated as embryonic stage 0.5 (E0.5). Cre activity was induced at E16.5 or E17.5 by a single intraperitoneal injection of Tamoxifen dissolved in sunflower oil/10% ethanol. Solutions ranging from 10–20 mg/ml were used with the following dosage on the basis of body weight: pregnant mothers of the *Shh-, Nfatc1-, Axin2-, Gli1-, Sox9-* and I*d2-CreER::Rosa-reporter* mice received a single dose of Tamoxifen totaling 40 µg/g body weight; the dose for the *Lgr5-GFP-CreER::Rosa-reporter* mice was 100 µg/g body weight.

### Whole-mount tail skin staining and imaging

At the indicated time points specified in figures, tail skin was removed and treated with 20 mM EDTA (pH 8.0) for 2–4 hr at 37°C. Epidermis with attached hair follicles was removed from the dermis, fixed in 4% paraformaldehyde for 20 min, washed in PBS for >2 hr and then processed for

staining or imaging. For the *Lgr5-GFP-CreER::Rosa-stop-tdTomato::K14H2BGFP* lineage tracing experiment, tail skin was removed and treated with 20 mM EDTA (pH 8.0) for 30 min at 37°C. Epidermis without hair follicles was removed from the dermis. The remaining dermal tissue with embedded hair follicles was fixed in 4% paraformaldehyde for 20 min, then washed for 1 hr in PBS and imaged without staining. For the mice more than 100 days old, the remaining dermal tissue was digested additionally in the collagenase (Sigma) for 1 hr and then washed with PBS before imaging to allow detection of fluorescence signal. Whole-mount tail skin images were acquired using a $20 \times 0.75$ objective lens. Z-stacks were acquired at a resolution of $1024 \times 1024$, or at $512 \times 512$ for large samples. Tissue and section samples were imaged with a Nikon A1-R confocal microscope. Microscopy data was analyzed using Imaris (3D software) with the 3D visualization module.

## Lineage tracing

To label distinct cell populations in hair germs at the chosen area of tail skin, pregnant mice were administered a single dose of Tamoxifen at E16.5, and the labeling pattern was established 48 hr later at E18.5 using embryos removed by Caesarian section. At this time point, the primary central follicle in tail skin follicle triplets is in the hair germ stage, while the secondary outer follicles have not yet been initiated. To continue lineage tracing, a foster mother nursed the remaining pups. At postnatal day 15 (P15), the initially labeled primary central follicles enter into the first telogen and the fates of the labeled cells were analyzed. At this stage the secondary outer follicles are in anagen and block the view of the telogen primary central follicle. Prior to imaging, the secondary outer hair follicles were plucked by tweezers, leaving only the central hair follicle in the P15 whole mount skin.

To label specific cell populations at the placode and the hair peg stages, pregnant mice were administered a single dose of Tamoxifen at E17.5 and labeling patterns were established 48 hr latter at P1 when the primary central follicle is at the hair peg stage and the secondary outer follicles are at the placode stage. The fate of labeled cell populations was traced to P15 for the primary central follicle and P21 for the secondary outer follicles, when they enter into the resting phase separately.

The development of tail skin hair follicles varies slightly, in both the anterior-posterior axis and in the dorsal-ventral axis. To label progenitor cells at defined developmental stages, only the middle one-third section along both the length and the width of dorsal tail skin was used to perform lineage-tracing experiments. Within this chosen area, hair follicle development consistently follows the above-mentioned pattern at the indicated time points.

To count the number of HFSCs, multiple Z-stack images were taken for each whole mount of tail skin. For each individual telogen HF, only the cross section through the center of the hair shaft was used for counting. This was done by picking the Z slice with the largest hair shaft for each counted HF. Telogen HFSC number was counted as the outer layer of cells from the U-shaped bulge structure below the SG and above the secondary hair germ. This standard was chosen based on section staining of tail skin hair follicles with HFSC markers (CD34 and Sox9) and niche cell marker (Krt6) (for reference, please see *Figure 5C–E* control HFs).

## In vivo imaging and laser ablation

P1 mice were anesthetized with cotton containing isoflurane and kept anesthetized on a 37°C heat stage during the imaging process with vaporized isoflurane through a gas tube connected to the head. The tail of the mouse was immobilized by tape and imaged directly under a water lens. Imaging was performed with a BX61WI (Olympus) microscope equipped with a Chameleon Ultra (COHERENT) two photon laser and a $25 \times$ water lens (Olympus, UIS2, N.A.1.05). A laser beam of 910 nm was used to simultaneously excite both H2BGFP and tdTomato. The step increment of the serial optical sections was 2 μm.

Prior to laser ablation, *Nfatc1-CreER::Rosa-stop-tdTomato::K14-H2BGFP* pregnant mice were injected with a single dose of Tamoxifen at E17.5 to label distinct cell populations in hair pegs on tail skin. Laser ablation was carried out with a 910 nm beam scanning a region of 10 μm$^2$ for less than 5 s. The power of the laser intensity was 1.86 mW, and less than 80% of the laser power was used. Neil Blue dye was used to tattoo a marker at the vicinity of the ablated area to help us retrieve the same laser ablated follicles more than 15 days later.

## Conditional knock out

To express the stabilized form of *β*-catenin in *Nfatc1* expressing cells, *Exon3-Ctnnb1^{fl/fl}::Rosa-stop-tdTomato^{fl/fl}* mice were mated with *Nfatc1-CreER* mice. To induce the conditional expression of *exon3-Ctnnb1*, nursing mothers of *Nfatc1-CreER::Rosa-stop-tdTomato^{fl/wt}::Exon3-Ctnnb1^{fl/wt}* pups and WT littermates were intraperitoneally injected with Tamoxifen doses totaling 40 µg/g body weight for 8 days starting from E17.5.

To generate *K14-rtTA::teto-Cre::Ctnnb1^{fl/fl}* mice, *K14-rtTA::teto-Cre* mice were mated with *Ctnnb1^{fl/fl}* mice. F1 male *K14-rtTA::teto-Cre::Ctnnb1^{fl/wt}* progeny were subsequently mated with female homozygous *Ctnnb1^{fl/fl}*. To induce deletion of *Ctnnb1*, the nursing mothers of *K14-rtTA::teto-Cre::Ctnnb1^{fl/fl}* pups and WT littermates were given water containing 1 mg/ml doxycycline (Sigma) and chow containing 2 g/1000 g doxycycline (Research Diets Company) for periods specified in the figures.

All mice were maintained in an SPF facility, and procedures were conducted in a manner consistent with the National Institute of Biological Sciences guide for the care and use of laboratory animals.

## Histology, immunofluorescence, confocal microscopy, and image processing

For section staining, tissues were embedded in OCT compound and frozen. After cryosection (20–30 µm) and fixed for 10 min in 4% paraformaldehyde in PBS, sections were permeabilized for 10 min in 0.5% Triton (PBST) and blocked for 1 hr in a solution of 2% normal donkey serum, 1% BSA, and 0.3% Triton in PBS. The following antibodies were used: anti-GFP (Abcam, ab13970, 1:1000), anti-P-cad (R&D, BAF761, 1:500), anti-β-gal (Abcam, ab9361, 1:10,000), anti-β-catenin (Sigma C2206, 1:500), anti-Sox9 (Chen Lab, 1:50), anti-CD34 (ebioscience, 50-0341, 1:300), and anti-K6 (Chen Lab, 1:2000).

For hematoxylin and eosin (H&E) staining, after cryosectioning (6–8 µm) and fixation for 10 min in 4% paraformaldehyde in PBS, sections were stained in haematoxylin (Fisher) for 30 s and then rinsed in running tap water and 0.3% acid alcohol. Eosin (Sigma) was used to treat the sections for 10 s. The haematoxylin and eosin dyes were prepared according to the manufacturers' instructions.

For the whole-mounts of tail skin, images were acquired using a 20× 0.75 objective lens. Z-stacks were acquired at a resolution of 1024×1024, or 512×512 for large samples. Tissue and section samples were imaged on a Nikon A1-R confocal microscope. Microscopy data were analyzed using Imaris (3D software) with the 3D visualization module. RBG images were assembled in Adobe Photoshop CS3 and panels were labeled with Adobe Illustrator CS6.

## Immunohistochemistry and in situ hybridization

Skin was fixed with 4% paraformaldehyde, embedded in paraffin, and sectioned at 5 µm. After dewaxing, sections were blocked with buffer from a MOM kit (Vector) with 0.1% Triton. Immunohistochemistry assays were then conducted by incubating sections at 4°C overnight with primary antibody β-catenin (Sigma C7207, 1:500), then secondary HRP Donkey-anti-mouse antibody (Jackson ImmunoResearch), followed by use of a DAB kit (Vector), according to the manufacturer's instructions.

RNA in situ hybridization was performed with an RNAscope 2-plex Detection Kit (Chromogenic) according to the manufacturer's instructions. The RNAscope probes used were Axin2 (Catalog 400331) and Wnt10b (Catalog 401071).

## Fluorescence-activated cell sorting (FACS)

Isolation of *Shh+* and *Nfatc1+* cells for RNA-seq was performed using *Shh-* and *Nfatc1-CreER::Rosa-stop-tdTomato::K14-H2BGFP* mice. For purification of *Nfatc1* expressing cells in hair pegs, pregnant mice were administered a single dose of Tamoxifen at E15.5. Then, at E17.5, embryonic back skin was removed and the whole skin was cut into 6–9 small pieces and placed, dermis side down, in a 0.25% trypsin solution (Gibco) at 37°C for 16 min. Single-cell suspensions were obtained by triturating the skin gently. Cells were then filtered with strainers (70 µm followed by 40 µm). For purification of *Shh*-expressing hair peg cells at E17.5, pregnant mice were administered a single dose of Tamoxifen at E15.5. Embryonic back skin was cut into 9 pieces and placed, dermis side down, in dispase

(Life Technologies, 0.4 mg/ml) dissolved in PBS for 1 hr at 37°C. Epidermis containing placodes and hair germs were removed. Dermis samples containing only hair pegs were placed in a 0.25% trypsin solution at 37°C for 10 min. Single-cell suspensions were obtained by triturating the skin gently. Cells were then filtered with strainers (70 μm followed by 40 μm).

For the cells used in the experiments presented in *Figure 5—figure supplement 1D*, back skin of *Nfatc1-CreER::Rosa-stop-tdTomato*$^{fl/wt}$*::K14-H2BGFP::Ctnnb1*$^{fl/wt}$ Het and WT mice at either P23 or P100 was used. Skin was cut into 6 to 9 pieces and placed, dermis side down, in a 0.25% trypsin solution at 37°C for 30 min. Epithelium cells were harvested by gentle scraping and triturating. After filtering with strainers, cells were stained for 20 min on ice with Alexa647-CD34 antibody (ebioscience, 50-0341) where indicated and then washed. DAPI was used to exclude dead cells. Cell isolations were performed with a BD FACSAria III cell sorter equipped with FACSDiva software (BD Bioscience).

## RNA isolation and real-time PCR

Total RNA was isolated from FACS-purified cells lysed with Trizol (Life Technologies) followed by extraction using a direct-Zol RNA mini prep kit (Zymo research). Equal amounts of RNA were added to a reverse-transcriptase reaction mix (Takara) with Oligo(dT). Expression levels were normalized to the expression of GAPDH or PPIB. Real-time PCR was conducted using a CFX96 Real-Time system (Bio-Rad) with Power SYBR Green PCR Master Mix (Life Technologies). All primer pairs were designed for the same cycling conditions, which were: 10 min at 95°C for initial denaturing, 40 cycles of 10 s at 95°C for denaturing, 30 s at 58°C for annealing, and 10 s at 65°C for extension. The primers were designed to produce a product spanning exon-intron boundaries in the target genes; the sequences were as follows:

Shh F, AAAGCTGACCCCTTTAGCCTA; Shh R, TTCGGAGTTTCTTGTGATCTTCC;
Nfatc1 F, CCATACGAGCTTCGGATCGA; Nfatc1 R, AGTAACCGTGTAGCTGCACAATG;
Wnt3 F, AGCTGCCAAGAGTGTATTCG; Wnt3 R, CTAGATCCTGCTTCTCATGGG;
Wnt10b F, AAGTCACAGAGTGGGTCAATG; Wnt10b R, GCCACGATAAACCCTAGACAG;
Lef1 F, AGCCTGTTTATCCCATCACG; Lef1 R, TGTTACAATAGCTGGATGAGGG;
PPIB F, GTGAGCGCTTCCCAGATGAGA; PPIB R, TGCCGGAGTCGACAATGATG;
GAPDH F, GGTGTGAACGGATTTGGCCGTATTG; GAPDH R, CCGTTGAATTTGCCGTGAGTGGAGT.

## RNA-seq analysis

RNA from FACS-purified cells was sent to the Biodynamic Optical Imaging Center at Peking University for quantification, RNA-seq library preparation, and sequencing. The libraries were sequenced on the illumina HiSeq 2500 platform using the Pair-End 2x100-bp sequencing strategy.

For analysis, data was mapped to the mouse genome (GRCm38/mm10) using TopHat (v2.0.13) with default settings. Cufflinks (v2.2.1) was used to quantify changes in gene expression between the *Nfatc1+* and *Shh+* cells. Genes with significantly different expression levels (p<0.01 and log2FC>1) were chosen for further analysis. Gene ontology (GO) analysis of genes was done using DAVID (Database for Annotation, Visualization and Integrated Discovery).

## Genotyping for the exon3-β-catenin mutant allele

DNA was isolated from FACS-purified cells using a TIANamp Genomic DNA Kit (TIANGEN) according to the manufacturer's instructions. The primers used to detect the β-catenin sequences flanking the exon3 region were: wild-type/un-recombined allele 900 bp; exon3 F, GACACCGCTGCGTGGACAATG; exon3 R, GTGGCTGACAGCAGCTTTTCT; recombined allele ~700 bp; exon3 F, GACACCGCTGCGTGGACAATG; exon3 R, ACGTGTGGCAAGTTCC GCGTCATCC.

## Statistics

Box-and-Whisker plots prepared with Origin 9 (Origin Lab) software were used to illustrate the lineage tracing quantification results.

## Acknowledgements

We are grateful to the NIBS Animal Facility for their expert handling and care of mice, the NIBS Biological Resource Centre for FACS sorting, the NIBS imaging facility for assistance with the two-photon microscope experiment, and the Peking University Biodynamic Optical Imaging Center for the RNA-seq sample processing. We thank all members of the Chen lab for discussions and technical support. We also thank N Tang, Y Zeng, R Xi, X Wang, F Shao, B Zhu, C Chang, and X Lim for discussion and manuscript reading. This work was supported by grants from the National Basic Research Program of China 973 Programs (2012CB518700 and 2014CB849602).

## Additional information

### Funding

| Funder | Grant reference number | Author |
|---|---|---|
| National Basic Research Program of China 973 Program | 2012CB518700 | Zijian Xu<br>Wenjie Wang<br>Kaiju Jiang<br>Zhou Yu<br>Huagnwei Huang<br>Fengchao Wang<br>Ting Chen |
| National Basic Research Program of China 973 Program | 2014CB849602 | Zijian Xu<br>Wenjie Wang<br>Kaiju Jiang<br>Zhou Yu<br>Huagnwei Huang<br>Fengchao Wang<br>Ting Chen |

The funders had no role in study design, data collection and interpretation, or the decision to submit the work for publication.

### Author contributions

ZX, TC, Conception and design, Acquisition of data, Analysis and interpretation of data, Drafting or revising the article, Contributed unpublished essential data or reagents; WW, KJ, Conception and design, Acquisition of data, Analysis and interpretation of data, Contributed unpublished essential data or reagents; ZY, Acquisition of data, Analysis and interpretation of data, Contributed unpublished essential data or reagents; HH, Analysis and interpretation of data, Contributed unpublished essential data or reagents; FW, BZ, Conception and design, Acquisition of data, Contributed unpublished essential data or reagents

### Ethics

Animal experimentation: This study was performed in strict accordance with the recommendations in the Guide for the Care and Use of Laboratory Animals of the National Institutes of Biological Sciences. All of the animals were handled according to the guidelines of the Chinese law regulating the usage of experimental animals and the protocols (M0020) approved by the Committee on the Ethics of Animal Experiments of the National Institute of Biological Sciences, Beijing. All surgery was performed under combination anesthesia involving isoflurane and remifentanyl and every effort was made to minimize discomfort and suffering.

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
