## [Decision Letter]

Thank you for submitting your work entitled "Embryonic attenuated Wnt/β-catenin signal defines niche location and long-term stem cells in hair follicle" for peer review at *eLife*. Your submission has been favorably evaluated by Janet Rossant (Senior editor), and two reviewers.

The reviewers have discussed the reviews with one another and the Reviewing editor has drafted this decision to help you prepare a revised submission.

Summary:

In this elegant and interesting study, Xu and colleagues ask a very important question, how are long term organ stem cells specified during development? Is hair follicle stem cell fate pre-determined at the early hair follicle developmental stages? Or, are they specified to be long term organ stem cells at different stages of development? They used multiple transgenic mice to trace cell lineage and found that Shh-expressing cells contribute to most of the HFSC, *Nfatc1*-expressing cells contribute to SG and upper bulge cells, and Lgr5-expressing cells contribute to lower bulge and SHG cells. By undertaking RNA-seq and using other transgenic lines, they also identified and demonstrated the attenuated Wnt/β-catenin signaling is required for HFSC specification.

The reviewers noted that this is a beautifully executed and extremely interesting paper that makes an important contribution to our understanding of adult stem cell development in the hair follicle, and likely has implications for understanding how adult stem cells are established in other biological systems as well. Therefore, the paper will be of interest to stem cell biologists in all fields. They raised a few major points, mostly around interpretation of the data presented and the need for some more caution in some of the conclusions drawn. It was felt that these concerns could probably be addressed without any further major experimentation.

Essential revisions:

1) The authors show that laser ablation of *Nfatc1-CreER*-labeled cells in embryogenesis did not prevent formation of a new bulge from unlabeled cells; the authors interpret this result as meaning that the bulge can be formed from *Nfatc1*-negative cells. However, in the absence of data showing that 100% of *Nfatc1*-expressing cells are labeled in *Nfatc1-CreER* mice following a single injection of tamoxifen, this conclusion should include a caveat regarding labeling efficiency.

And related to this point, in Figure 1, the authors mention that *Nfatc1*-expressing progenitor cells become SG cells and upper bulge cells. In Figure 2, the authors claim that the formed HFSCs are negative for *Nfatc1* at P23 after *Nfatc1* ablation at P1, indicating they came from hair peg cells that used to be *Nfatc1* negative. But the formed SG cells are *Nfatc1* positive in Figure 2 but negative in Figure 2, which is inconsistent. If Figure 2 is correct, where do these cells come from?

2) To understand the mechanisms underlying the different developmental potential of Shh-expressing and *Nfatc1*-expressing cells, the authors carried out transcriptional profiling on these two populations, and identified Wnt/β-catenin signaling as a pathway enriched in Shh-expressing and downregulated in *Nfatc1*-expressing cells. Lineage tracing with *Axin2-CreER* mice confirmed that while *Axin2*-expressing cells labeled at the hair placode stage can give rise to HFSCs, induction of Cre activity at the hair peg stage resulted in labeling of only a small percentage of HFSCs. The authors conclude that Wnt signaling attenuates the ability of cells to become HFSCs. In support of this, they demonstrate loss of HFSC markers in hair follicles following mutation of epithelial β-catenin to a constitutively active form. Conversely, deletion of epithelial β-catenin in late embryonic life causes expanded expression of the HFSC marker Sox9. While these data are consistent with the authors' hypothesis, it also remains possible that a minority of *Axin2*-positive (Wnt-responsive) cells, located in the upper follicle at the peg stage, does in fact contribute to HFSC rather than differentiating HF populations, and is distinguished from differentiating cells by lower levels and/or a different context of Wnt signaling. As it isn't clear that Lgr5+ and *Axin2+* cells represent identical (rather than overlapping) populations, this alternate hypothesis could be tested by determining whether *Axin2*-lineage traced cells in the bulge compartment (Figure 4) are maintained and/or expand, or conversely disappear, through successive cycles of hair growth.

3) The authors should consider further whether the conclusion at the end of the second paragraph of the subsection “Attenuated Wnt/β-catenin signal uniquely associates with embryonic niche “is correct or not. From placode to peg stage, a lot of proliferation, differentiation and other cellular events are involved. Is true that a specific group of cells can maintain active Wnt/β-catenin and then loss HFSC forming potential?

The authors should consider whether Wnt signaling is a cause or a consequence of the differentiation process and modify their conclusions accordingly.

4) By using Shh, *Nfatc1* and *Lgr5-Cre* lines to undertake lineage tracing from early development to the end of morphogenesis, the authors claim that HFSCs are specified at the hair peg stage. However, it is probably appropriate to conclude that HFSCs are derived from those progenitor cells, but not they are already specified at an early developmental stage.

---

## [Author Response]

Essential revisions:

1) The authors show that laser ablation of Nfatc1-CreER-labeled cells in embryogenesis did not prevent formation of a new bulge from unlabeled cells; the authors interpret this result as meaning that the bulge can be formed from Nfatc1-negative cells. However, in the absence of data showing that 100% of Nfatc1-expressing cells are labeled in Nfatc1-CreER mice following a single injection of tamoxifen, this conclusion should include a caveat regarding labeling efficiency.

We appreciate the reviewers pointing out the caveat of the ablation experiment. We agree with the concerns and would like to first explain the experiment more fully. In Figure 1: the lineage tracing experiments used the dorsal middle 1/3 section of the tail whole mount for quantification (as explained in Figure 1). Although hair follicles in this chosen region all follow the developmental pattern we described, we did observe slight variation in labeling efficiency after one Tamoxifen injection (please see added Figure 2—figure supplement 1). Specifically, hair follicles along the anterior-posterior midline of dorsal tail skin had the highest labeling efficiency compared to hair follicles on the sides. For this reason, we only conducted the ablation experiment in hair follicles along this midline in the chosen region. To illustrate our point please see Figure 2: control hair follicles surrounding the ablated hair follicles have almost all of their bulge cells labeled after just one Tamoxifen injection at E17.5. But as the reviewers pointed out, even in this situation, we still cannot prove that we labeled and then ablated all of the cells that were *Nfatc1* positive. Therefore, we have modified our conclusions accordingly based on the reviewers’ suggestions. Relevant labeling is also changed in Figure 2.

And related to this point, in Figure 1, the authors mention that Nfatc1-expressing progenitor cells become SG cells and upper bulge cells. In Figure 2, the authors claim that the formed HFSCs are negative for Nfatc1 at P23 after Nfatc1 ablation at P1, indicating they came from hair peg cells that used to be Nfatc1 negative. But the formed SG cells are Nfatc1 positive in Figure 2 but negative in Figure 2, which is inconsistent. If Figure 2 is correct, where do these cells come from?

In Figure.2C, the labeled SG cells resulted from incomplete ablation of the *Nfatc1-CreER* labeled cells at the hair peg stage. Even though we did aim to specifically ablate all RFP positive cells in hair pegs, in a very few cases, we failed to fully achieve this. This was revealed by remnant RFP positive cells upon later observation. However, in the majority of our repeat experiments, we did successfully ablate all *Nfatc1-CreER* labeled hair peg cells, which resulted in neither SG nor bulges being labeled at later time points (as indicated by Figure 2). We have changed the representative images in Figure 2 to more accurately reflect the results of the majority of the ablation experiments.

2) To understand the mechanisms underlying the different developmental potential of Shh-expressing and Nfatc1-expressing cells, the authors carried out transcriptional profiling on these two populations, and identified Wnt/β-catenin signaling as a pathway enriched in Shh-expressing and downregulated in Nfatc1-expressing cells. Lineage tracing with Axin2-CreER mice confirmed that while Axin2-expressing cells labeled at the hair placode stage can give rise to HFSCs, induction of Cre activity at the hair peg stage resulted in labeling of only a small percentage of HFSCs. The authors conclude that Wnt signaling attenuates the ability of cells to become HFSCs. In support of this, they demonstrate loss of HFSC markers in hair follicles following mutation of epithelial β-catenin to a constitutively active form. Conversely, deletion of epithelial β-catenin in late embryonic life causes expanded expression of the HFSC marker Sox9. While these data are consistent with the authors' hypothesis, it also remains possible that a minority of Axin2-positive (Wnt-responsive) cells, located in the upper follicle at the peg stage, does in fact contribute to HFSC rather than differentiating HF populations, and is distinguished from differentiating cells by lower levels and/or a different context of Wnt signaling. As it isn't clear that Lgr5+ and Axin2+ cells represent identical (rather than overlapping) populations, this alternate hypothesis could be tested by determining whether Axin2-lineage traced cells in the bulge compartment (Figure 4) are maintained and/or expand, or conversely disappear, through successive cycles of hair growth.

We agree with the reviewers’ point. We are also interested in finding out the fate of the *Axin2* positive cells located in the upper follicle that ended up in the bulge: are they long-term stem cells or not? Since it is not clear that Lgr5+ and *Axin2+* cell are the same, we followed the fate of *Axin2+* cells in long term lineage tracing experiments (please see new Figure 6). After injecting Tamoxifen once at E17.5 to label *Axin2+* hair peg cells, we got consistent labeling efficiency of the first telogen bulge (~15%). However, following just two rounds of successive hair cycles, most of the *Axin2+* cells contributed bulge stem cells disappeared (please see new Figure 6). This indicates that although a few Wnt signaling responsive cells in the hair peg participate in bulge formation, they are not long-term stem cells and tend to exhaust after participating in downstream proliferation and differentiation. These results suggest that embryonic exposure to Wnt/β-catenin signaling diminishes the long-term potential of adult HFSCs.

3) The authors should consider further whether the conclusion at the end of the second paragraph of the subsection “Attenuated Wnt/β-catenin signal uniquely associates with embryonic niche “is correct or not. From placode to peg stage, a lot of proliferation, differentiation and other cellular events are involved. Is true that a specific group of cells can maintain active Wnt/β-catenin and then loss HFSC forming potential?

The authors should consider whether Wnt signaling is a cause or a consequence of the differentiation process and modify their conclusions accordingly.

We agree that the statement at the end of the second paragraph of the subsection “Attenuated Wnt/β-catenin signal uniquely associates with embryonic niche “was not fully supported by the data up to that point. Therefore, we have modified the conclusions according to the reviewers’ suggestions.

Although we don’t have direct evidence to distinguish whether Wnt signaling is a cause or a consequence of the differentiation process, in our K14rtTA,tetoCre,β-catenin cKO animals, hair coat formation was aborted (see Figure 5; Figure 5—figure supplement 1). This indicates at least that epithelium Wnt/β-catenin signaling is required for terminal differentiation and hair formation. That being said, we do appreciate the reviewers’ suggestions that we should consider further and be more cautious when drawing conclusions. We have modified multiple statements in the paper to reflect this suggestion.

4) By using Shh, Nfatc1 and Lgr5-Cre lines to undertake lineage tracing from early development to the end of morphogenesis, the authors claim that HFSCs are specified at the hair peg stage. However, it is probably appropriate to conclude that HFSCs are derived from those progenitor cells, but not they are already specified at an early developmental stage.

We agree. It is more appropriate to conclude that HFSCs are derived from those progenitor cells. We have changed the corresponding conclusions based on this suggestion as well as multiple relevant statements in the paper.